# Improving Polyphenolic Compounds: Antioxidant Activity in Chickpea Sprouts through Elicitation with Hydrogen Peroxide

**DOI:** 10.3390/foods9121791

**Published:** 2020-12-02

**Authors:** Liliana León-López, Yudith Escobar-Zúñiga, Nancy Yareli Salazar-Salas, Saraid Mora Rochín, Edith Oliva Cuevas-Rodríguez, Cuauhtémoc Reyes-Moreno, Jorge Milán-Carrillo

**Affiliations:** 1Programa de Posgrado Integral en Biotecnología, Facultad de Ciencias Químico Biológicas, Universidad Autónoma de Sinaloa, Culiacán, Sinaloa C.P. 80000, Mexico; lili.leon@uas.edu.mx (L.L.-L.); nancy.salazar@uas.edu.mx (N.Y.S.-S.); smora@uas.edu.mx (S.M.R.); edith.cuevas.r@uas.edu.mx (E.O.C.-R.); creyes@uas.edu.mx (C.R.-M.); 2Programa de Posgrado Integral en Ciencia y Tecnología de Alimentos, Facultad de Ciencias Químico Biológicas, Universidad Autónoma de Sinaloa, Culiacán, Sinaloa C.P. 80000, Mexico; yudi.1412@hotmail.com

**Keywords:** chickpea sprouts, elicitation, hydrogen peroxide, response surface methodology, polyphenolic compounds, antioxidant activity

## Abstract

Elicitation appears to be a promising alternative to enhance the bioactive compound content and biological activities of legume sprouts. Multi-response optimization by response surface methodology (RSM) with desirability function (DF) was used to optimize the elicitor concentration (hydrogen peroxide (H_2_O_2_)) and germination time in order to maximize total phenolic content (TPC), total flavonoids content (TFC), and antioxidant activity (AOX) of chickpea sprouts. Chemical, antinutritional, and nutraceutical properties of optimized chickpea sprouts (OCS) were also determined. The predicted regression models developed were efficiently fitted to the experimental data. The results of the desirability function revealed that optimum attributes in chickpea sprouts can be achieved by the application of 30 mM H_2_O_2_ and 72 h of germination time, with global desirability value D = 0.893. These OCS had higher (*p* < 0.05) TPC (7.4%), total iso-flavonoids (16.5%), AOX (14.8%), and lower phytic acid (16.1%) and saponins (21.8%) compared to H_2_O_2_ non-treated chickpea sprouts. Optimized germination conditions slightly modified the flavonoid profile in chickpea; eight iso-flavonoids were identified in OCS, including formononetin and biochanin A, which were identified as the major compounds. Results from this study support elicitation with H_2_O_2_ as an effective approach to improve phytochemical content and antioxidant activity in chickpea sprouts.

## 1. Introduction

Chickpea (*Cicer arietinum* L.) is an important legume extensively consumed around the world. This pulse crop varies in size, shape, and color and is broadly cultivated in the Northwest of Mexico [1]. Furthermore, chickpea contains a rich basis of several healthy constituents, such as proteins, starch, fiber, minerals, vitamins, health-beneficial fatty acids, and phenolic compounds, for instance, phenolic acids and flavonoids, mainly iso-flavonoids [2,3,4]. Legume iso-flavonoids have received considerable attention for exhibited protective effects against cardiovascular disease, osteoporosis, menopausal symptoms, and hormone-dependent cancer [5,6].

Germination (sprouting) is an affordable and efficient strategy for processing legume seeds to develop their nutritional quality and obtain sprouts with nutraceutical potential [7,8]. Several studies have established that accumulation of a diverse range of bioactive compounds increases during germination in seeds. These bioactive compounds have been documented to have antioxidant, anticarcinogenic, antidiabetic, hypolipidemic, and anti-inflammatory actions [8,9,10]. Therefore, edible sprouts could be considered foods with potential health-promoting properties [11,12]. Overall, germination conditions can affect the synthesis and content of bioactive compounds in the seeds. Consequently, many studies have been focused on strategies for increasing the metabolite production in seed sprouts.

Nowadays, there has been developing interest in applying exogenous elicitors during the germination of seeds to stimulate seedling growth and increase their bioactive compounds [13,14]. Elicitors induce oxidative stress by increasing reactive oxygen species (ROS) and reactive nitrogen species (RNS) production. As a protective mechanism against the damage caused by this ROS and RNS imbalance, plants have developed an antioxidant defense system that includes enzymatic and non-enzymatic response. The non-enzymatic response involves the biosynthesis of antioxidant molecules such as phenolic compounds [15,16,17]. Previous studies have shown that the application of elicitors such as temperature, salinity, and exposure to chemical substances, including hydrogen peroxide (H_2_O_2_) during legume sprouting, induces the accumulation of secondary metabolites with biological action [18,19,20,21,22].

Some researchers have reported the effect of different elicitors on the accumulation of phenolic compounds in chickpea sprouts [3,6,23]. However, little has been reported about the effect of elicitation with H_2_O_2_ on the chemical, antinutritional, and nutraceutical quality of chickpea sprouts. Therefore, the current research aimed to optimize H_2_O_2_ concentration and germination time to obtain chickpea sprouts with enhanced phytochemical content (total phenolic content (TPC) and total flavonoids content (TFC) and antioxidant activity (AOX) by the application of response surface methodology (RSM) and DF, and the influence of the optimized germination states on the antinutritional and nutraceutical properties from the chickpea sprouts.

## 2. Materials and Methods

### 2.1. Materials

The study was carried out on chickpea (*Cicer arietinum* L.) cultivar Blanco Sinaloa 92 grown and harvested in 2018 in Angostura, Sinaloa, Mexico. The kernels were cleaned, kept in bottles and stored at a cold temperature (−18 °C) until further examination.

### 2.2. Chickpea Germination Process

According to the protocol described by Guardado-Félix et al. [3], the germination process was carried out, with some modifications. Chickpea seeds (CS) were sanitized in 0.5% sodium hypochlorite solution for 10 min and then rinsed three times through purified water to remove the sanitizing solution. Afterward, elicitation treatments were applied on chickpea seeds during the soaking (imbibition) step. Batches of 100 g of seeds were soaked with different hydrogen peroxide concentrations (according to the experimental design, Table 1) for 24 h at room temperature. After the soaking treatment, the soaking solution was rinsed with distilled water. The elicited hydrated seeds were spread uniformly into a container (40 × 30 cm), and the germination process was conducted in an incubator chamber (25 °C and 80–90% of relative humidity) until completion of the programmed germination time (Table 1). After germination, the germinated chickpea (GC) was freeze-dried, the lyophilized samples were milled to pass through an US 80 sieve, and then the resulting flours were kept in plastic bags at −20 °C for further examination.

### 2.3. Germination Percentage

The germination percentage (GP) was calculated based on the total quantity of sprouts completely developed. The germination percentage results were estimated from the following equation:$$\mathrm{GP}=\left[\frac{GC_{seeds}}{\mathrm{Total}\;CS_{seeds}}\right]\times 100$$
where *GC_seeds_* represents the number of germinated seeds. Seeds with visible radicles were considered as germinated. *CS_seeds_* represents the total number of seeds treated.

### 2.4. Preparation of Phytochemical Extracts

The extracts were obtained from the raw and germinated samples of chickpea. The samples were extracted with acetone/ethanol/water (40:40:20, *v*:*v*), agreeing to Milán-Noris et al. [4]. The mixture was later centrifuged at 4000× *g* for 10 min, and the supernatant was collected and reserved at −20 °C as far as future evaluation.

### 2.5. Determination of Total Phenolic Content and Total Flavonoids Content

The Folin-Ciocalteu method was implemented to determine the total phenolic content (TPC), according to Singleton et al. [24]. Results were expressed as milligrams of gallic acid equivalent (GAE) per 100 g of the dry weight basis (mg GAE/100 g DW). The total flavonoids content (TFC) was determined by Heimler et al. [25]. Results were represented as milligrams of catechin equivalent (CE) per 100 g of the dry weight basis (mg CE/100 g DW).

### 2.6. Antioxidant Activity

The ABTS radical scavenging assay was conducted as described to the methodology explained by Re et al. [26]. The ABTS value was described as micromoles of Trolox equivalent (TE) per 100 g of the dry weight basis (µmol TE/100 g DW).

### 2.7. Identification and Quantification of Flavonoids

Flavonoids in chickpea extracts were identified and quantified, according to Guardado-Félix et al. [3] with slight modifications. A 10 μL aliquot of the methanol extract was injected into a UPLC-DAD system (ACCELA, Thermo Scientific, Waltham, MA, USA). The separation was performed in a C18 column (3 μm, 50 × 2.1 mm) (Fortis Technologies Ltd., Cheshire, UK) using as mobile phases 1% (*v*/*v*) formic acid (A) and 100% (*v*/*v*) acetonitrile (B) and a linear gradient from 0.5 to 60% of B in 35 min at 0.2 mL/min. The chromatograms were registered at 260 and 295 nm. The UPLC-DAD was connected to a mass spectrometer with an electrospray ionization source (ESI) (LTQ XL, Thermo Scientific, San Jose, CA, USA) operating in positive/negative mode (35 V, 300 °C). Data were analyzed with the Xcalibur 2.2 software (Thermo Scientific, San Jose, CA, USA), and full scan spectra were acquired in the *m/z* range of 110 to 2000. Selected ions for MS experiments were fragmented by collision induced dissociation applying 10–45 V. Helium and nitrogen were used for collision and drying, respectively. Flavonoids identification was based on UV-spectra and by comparison with mass spectrometry (MS) fragmentation patterns of compounds previously identified in the literature, and MS data obtained from commercial standards biochanin A and formononetin (Sigma Aldrich, St. Louis, MO, USA). The total iso-flavonoids content was quantified from the calibration curve of formononetin; the results were expressed as micrograms of formononetin equivalent (FE) per 100 g of the dry weight basis (μg FE/100 g DW).

### 2.8. Chemical Composition and Antinutritional Components

#### 2.8.1. Chemical Composition

The moisture (method 925.09), protein (method 920.87), ash (method 923.03), and lipid (method 922.6) of chickpea sprouts were assessed following AOAC official methods [27]. The carbohydrates content was calculated by the difference of moisture, protein, fat, and ash from 100 g of chickpea sprouts. Results were expressed as % dry weight basis (DW).

#### 2.8.2. Phytic Acid

Phytic acid (PA) content was established by the colorimetric procedure explained by Gao et al. [28]. Absorbances was measured at 500 nm using a spectrophotometer (Victor^TM^ X3 Multi-Label, Perkin Elmer, Inc., Hopkinton, MA, USA), and PA content was assessed utilizing calibration curves with the phytic acid standard. Outcomes were described as milligrams of PA per 100 g of dry weight basis (mg PAE/100 g DW).

#### 2.8.3. Saponin Content

Saponin content was estimated following the spectrophotometric methodology designated by Dini et al. [29]. The dried methanolic extracts were prepared with 80% methanol, and absorbance was measured at 520 nm through a spectrophotometer (Victor^TM^ X3 Multi-Label, Perkin Elmer, Inc., Hopkinton, MA, USA). The concentration of saponins was taken from a standard curve of diosgenin. Outcomes were defined as milligrams of Diosgenin equivalents (DE) per 100 g of the dry weight basis (mg DE/100 g DW).

#### 2.8.4. Trypsin Inhibitors Activity

The trypsin inhibitor activity (TIA) in chickpea samples was appraised to the enzymatic assay described by Domoney & Welham [30] using Na-Benzoyl-L-arginine 4-nitroanilide hydrochloride (BAPNA) as trypsin substrate. Absorbance was measured at 410 nm using a spectrophotometer (Victor^TM^ X3 Multi-Label, Perkin Elmer, Inc., Hopkinton, MA, USA), and TIA was reported as trypsin inhibitor units (TIU) per milligram of dry weight basis (TIU/mg DW).

### 2.9. Regression Analysis and Optimization

The optimization process was carried out using RSM for establishing the optimal germination conditions (elicitor concentration (H_2_O_2_) and germination time) to obtain chickpea sprouts. The process variables of this study were H_2_O_2_ soaking concentration (X_1_ = (H_2_O_2_), 5–30 mM) and germination time (X_2_ = Gt, 12–72 h), while the dependent variables, TPC, TFC, and AOX were selected as the responses measured. Data from preliminary trials chose the amount and array of process variables in the experimental design. Table 1 exhibited a central composite design (CCD) containing 13 experiments developed in random order. The response variables were fitted by the quadratic polynomial model equation as follow:$$Y=\beta_{O}+\overset{2}{\underset{i=1}{\sum }}\beta_{i}X_{i}+\overset{2}{\underset{i=2}{\sum }}\beta_{ii}X_{i}^{2}+\overset{2}{\underset{i=1}{\sum }}\overset{2}{\underset{i=j+1}{\sum }}\beta_{ij}X_{i}X_{j}+\epsilon$$
where *Y* is the predicted response variable; while *β_0_*, *β_i_*, *β_ii_*, and *β_ij_* are regression coefficients for the intercept, linear, quadratic, and interaction terms, respectively, and *X_i_* and *X_j_* represent the independent variables. The polynomial coefficients were calculated and analyzed using the Design Expert (Version 12) software. Data were subjected to stepwise regression analysis; the significant terms (*p* < 0.05) were used to fit the predictive model for each response variable [31]. According to the software, multiple response optimization was implemented throughout the desirability function [32]. To establish the desirability of several arrangements of the experimental process, variables ([H_2_O_2_] and Gt) were set as ‘‘in the range,” whereas that of response variables (TPC, TFC, and AOX) was set as a goal to obtain maximum, and the desirability value was calculated. The arrangement of experimental factors yielding the uppermost desirability was nominated as the optimal germination condition.

### 2.10. Statistical Analysis

Chemical composition, antinutritional parameters and bioactive compounds of unprocessed samples, and chickpea sprouts were subjected to a one-way analysis of variance (ANOVA), and the comparison of means was carried out with Duncan’s multiple range test with a confidence level of 95% (*p* < 0.05) using Statgraphics Centurion XVI software (Statistical Graphics Corporation, Rockville, MD, USA).

## 3. Results and Discussion

### 3.1. Percentage of Germination

Figure 1 exhibits the outcome of elicitor (H_2_O_2_) on the germination percentage of chickpea seeds. All H_2_O_2_ concentrations assayed had a positive effect on chickpea germination even at 0 h after imbibition. Many facts could explain early germination induction; H_2_O_2_ helps crack hard seeds, allowing water interaction. H_2_O_2_ can also oxidize compounds in the pericarp that act as germination inhibitors such as alkaloids [33,34]. On the other hand, it has been demonstrated that H_2_O_2_ interferes with hormonal barriers that inhibit germination [34].

Our results showed that, between 24 and 72 h of germination time, the germination percentage reached the highest level (80.0–91.1%) when 5 to 30 mM H_2_O_2_ concentrations were applied. The 5 mM H_2_O_2_ concentration induced the best response based only on the germination percentage. Interestingly, H_2_O_2_ concentration above 30 mM significantly reduced the percentage of germination in chickpea (66.7–72.2%), whereas the germination percentage of the seeds treated with 0 mM H_2_O_2_ remained at 41.11%. The results of this study agreed with by Barba-Espin et al. [15], which reported that elicitation with H_2_O_2_ on pea seeds promoted germination and affected germinative performance, increasing the germination percentage

Regarding the effect of germination time on germination percentage, the highest values reached after 48 h of germination time in all treatments. Sohail et al. [35] observed a higher germination percentage in non-treated chickpea than observed in this study (76%). These authors also assayed the effect of potassium chloride and polyethylene glycol (PEG) on chickpea germination parameters. They reported the highest percentage of germination when PEG 2% was used (88%), followed by the PEG 1% treatment (84.3%). However, the germination time used was not declared.

### 3.2. Appropriate Models by Response Variables

The quadratic polynomial equations for each response variable (TPC, TFC, and AOX) were efficiently fitted to the experimental values of germination conditions ([H_2_O_2_] and Gt) using multiple regression analysis (Table 1). According to Vera-Candiotia et al. [36], the validity of the appropriate and reproducible predictive model with high accuracy should be confirmed considering the following statistical parameters: coefficient of determination (R^2^) and adjusted-R^2^ high (>0.80), very small *p*-value (<0.05), coefficient of variation (CV < 10%), adequate precision (PRESS) > 4, and lack of fit test (*p* > 0.0.5). In this study, the results reveal that the regression analysis for each response model was significant (*p* < 0.001), with a coefficient of determination (R^2^), CV, and PRESS in the range of 0.0.933–0.967, 4.8–8.3% and PRES 14.1–23.8, respectively. These models did not present a lack of fit (*p* > 0.05) (Table 2). These statistical parameters indicated that the experimental data of the response variables studied correlated positively with the quadratic polynomial models selected in the experimental design.

#### 3.2.1. Total Phenolics Content (TPC)

Phenolic compounds are the most prominent group of bioactive compounds present in legume sprouts [8]. The regression analysis showed that linear terms H_2_O_2_ concentration (X_1_) and germination time (X_2_), quadratic term germination time $(X_{2}^{2}$), and interaction term between H_2_O_2_ concentration/germination time (X_1×2_) had significant (*p* < 0.05) effects on TPC of chickpea sprouts. The statistical parameters demonstrated that the fitted model was suitable and reproducible (Table 2). The surface response plots are revealed in Figure 2A. The highest amounts of TPC (140.1 mg GAE/100 g DW) were observed at (H_2_O_2_) = 20.0–35.0 mM and Gt = 78.0–84.0 h, indicating that increase in [H_2_O_2_] and Gt were necessary for maximizing the TPC of chickpea sprouts. These results coincide with early research reporting a time-dependent phenolic compounds enlargement throughout germination in chickpea seed [37]. The increase of phenolic compounds in chickpea sprouts treated by elicitor (H_2_O_2_) could be attributed to the induction of phenylalanine ammonia-lyase (PAL) enzyme involved in the biosynthesis of phenolic compounds [38].

#### 3.2.2. Total Flavonoids Content (TFC)

Flavonoids are the most outstanding collection of phenolic composites presented in legumes. Changes in TFC of the chickpea sprouts were significantly (*p* < 0.05) affected by the linear (X_1_ and X_2_) and the quadratic ($X_{1}^{2}$ and $X_{2}^{2}$) terms. The regression developed model described 96.7% of the total variability to TFC on chickpea sprouts, suggesting that on this examination the selected model illustrates the information adequately for this response (Table 2). Total flavonoids were increased by the effect of (H_2_O_2_) and germination time. The highest TFC (30.0 mg CE/100 g DW) was found in sprouts treated with 20 to 35 mM H_2_O_2_ and 80 to 84 h, respectively (Figure 2B). The increase in TFC found in chickpea sprouts could be related to the induction of PAL, and other enzymes involved in the phenylpropanoid biosynthesis pathway by the effect of H_2_O_2_ application, triggering the accumulation of these secondary metabolites [39].

#### 3.2.3. Antioxidant Activity (AOX)

Antioxidant activity is the most widely examined bioactivity in germinated edible seeds and sprouts [8]. The regression analysis shows that AOX obtained by ABTS assay were significantly (*p* < 0.05) dependent on the linear terms (**X_1_** and **X_2_**) and the interaction term (**X_1_X_2_**). The regression model explained 93.3% of the total variability to AOX on chickpea sprouts (Table 2). The AOX found in chickpea sprouts depended on H_2_O_2_ concentration and germination time (Figure 2C). Among all of the sprouted chickpea tested, [H_2_O_2_] (=27.5–35.0 mM/Gt = 75.0–84.0 h revealed the highest AOX (2345.4 mg TE/100 g DW). Some researchers suggest that germination treatments with elicitor in legumes led to the increment of phenolic compounds, increasing the antioxidant activities of the sprouts [18,40].

### 3.3. Optimization and Validation of Germination Conditions

Desirability optimization of the germination condition was carried out for accumulative phenolic compounds and AOX from chickpea seed. During desirability determination, the criteria proposed for selecting the optimum conditions for chickpea sprouts were for independent variables; [H_2_O_2_] and Gt were kept in range, and responses TPC, TFC, and AOX were maximized. The optimization procedure was conducted under these conditions and restrictions. Figure 3 exhibited the desirability function’s response surface to attaining optimum conditions in chickpea sprouts. The applying the desirability function method suggests that the optimal germination conditions for elaboration of optimized chickpea sprouts (OCS) were those that create the highest global desirability value (D = 0.893), represented when the H_2_O_2_ concentration and germination time was 30 mM and 72 h, respectively.

The established regression models’ fitness was further validated by comparing the experimental values with those predicted by the models under optimum conditions. In this study, the values obtaining experimentally for all response variables (TPC, TFC, AOX) on the OCS were 138.3 mg GAE/100 g DW, 27.8 mg CE/100 g DW, and 2347.2 µmol TE/100 g DW, respectively, which were closely matched with theoretically predicted values (Table 2).

The results denote the correctness of the established quadratic models, and it can be noted that these best values are acceptable inside the specified array of process factors. Hence, this technique is still convenient for optimization, particularly when used in a mixture with other procedures.

### 3.4. Chemical Composition, Antinutritional and Nutraceutical Properties of Optimized Germinated Sprouts

The chemical composition values attained from optimized chickpea sprouts (OCS) were protein content (18.93%), lipids (7.96%), ash (2.77%), and carbohydrates (71.34%) (Table 3). Throughout germination, lipids, ash, and carbohydrate contents were commonly degraded to provide energy by kernel development, thereby decreasing its content. However, these results denote that the chemical stress induced by the H_2_O_2_ elicitor used in the present study did not improve chemical composition values when comparing OCS and control chickpea sprouts (Table 3).

The presence of antinutritional factors in edible legumes reduces the nutritional value of consumed food. Therefore, the inactivation or removal of these undesirable components through germination and elicitation may improve the nutritional quality of legumes [41]. There are numerous reports on the effect of elicitors on antinutritional compounds of sprouts. Some elicitors can decrease these under appropriate conditions [18,42].

Phytates have been considered unfavorable in foods due to their ability to decrease the bioavailability of minerals such as Zn, Ca, K, Mg, and Mn, which influence their nutritional attributes [43]. The phytic acid content in OCS was significantly reduced by optimal germination conditions (30.6%) compared to unprocessed chickpea. Comparable results have been described earlier in common bean sprouts, where chemical elicitors such as H_2_O_2_, chitosan, and salicylic acid caused a significant reduction in phytic acid content [18]. The reduction of the phytic acid content detected in OCS can be attributed to the hydrolyzation of inositol hexa-phosphate to lower molecular weight forms during the germination [44]. On the other hand, optimal germination conditions did not reduce significantly (*p* > 0.05) the trypsin inhibitor activity in chickpea sprouts (Table 3). However, in contrast to our results, some researchers, Mendoza-Sánchez et al. [18] and Swieca & Baraniak [40], revealed that treatment with H_2_O_2_ favored the reduction of trypsin inhibitor activity in legume sprouts.

Saponins are a complex and diverse group of compounds, and recently, their adverse effects and health-promoting properties have been controversially studied [45,46]. The germination process caused an increase in levels of saponin by 16.9% compared to unprocessed chickpea. Nevertheless, the elicitation germination condition produced a significant reduction in the saponin content of OCS by 21.8% compared to H_2_O_2_ non-treated chickpea sprouts (Table 3). Several studies report an increase of saponin content in legume sprouts. For instance, seed treatment with H_2_O_2_ caused an increase (55%) of saponin content in common bean sprouts compared to control [18]. Guajardo-Flores et al. [46] reported these saponin content changes in sprouting black beans, which increased up to 1.9-fold compared to raw seeds. Overall, the effects of elicitation with H_2_O_2_ in chickpea sprouts on antinutritional parameters are inconsistent. The reported changing tendency results could be attributed to the comprehensive effect of many factors, such as genetic characteristics of sprouts, growth phase, the growing environment of sprouts, and elicitor nature [16].

Physiological changes can be induced in sprouting seeds treated with elicitors; the seeds respond to stresses from the stimulating molecular signaling pathways, which lead to the enhanced production and accumulation of secondary metabolites [35,36]. Bioprocessing by germination treated with H_2_O_2_ increased the TPC and TFC during OCS elaboration. Both TPC and TFC were increased significantly (3.0 and 2.7-fold, respectively) compared to unprocessed grains (Table 3). These results agree with the same tendency observed in lentil sprouts treated with H_2_O_2_ [14] and common bean sprouts after H_2_O_2_ application [18]. The increase in phenolics compounds found in OCS could be explained by the fact that elicitors stimulate the biosynthetic pathway of phenolic compounds triggering their accumulation in sprouts [38].

Figure 4 exhibit the flavonoid profiles on phytochemical extracts obtained from OCS. The identification of flavonoid compounds was recognized by comparing their spectroscopic and chromatographic characteristics with those of the standards. Those compounds without commercially available standards were tentatively identified using retention time arrangement, peak spectra, mass-to-charge ratio, MS fragmentation, and online metabolite databases, including PubChem and Scifinder Scholar^TM^ [47]. In our study, 10 flavonoids were detected and identified in OCS, and the majority of them are iso-flavonoids: formononetin glucoside, formononetin glucoside malonylated, biochanin A glucoside, calicosin, prunetin, biochanin A, pseudo-baptigenin, and formononetin. Besides, one flavanonol (garbanzol) and one saponin (soysaponin βg II) were also identified (Figure 4). As exhibited in Figure 4, the principal compounds in chickpea sprouts are formononetin (peak 9), formononetin glucoside (peak 2), formononetin glucoside malonylated (peak 3), biochanin A glucoside (peak 4), and prunetin (peak 6). The isoflavones biochanin A and formononetin were the most abundant in the OCS (Figure 4). These iso-flavonoids are similar to those previously reported in chickpea sprouts by Wu et al. [47].

Otherwise, the highest total iso-flavonoid content was obtained when the optimal germination condition was applied to chickpea seeds (Table 3). A concentration of 257.9 µg FE/100 g DW was obtained for OCS. In contrast, unprocessed chickpea seed contained only 8.5 µg FE/100 g DW (Table 3). In the presented study, chickpea sprouts elicited with 30mM H_2_O_2_ and 72 h of germination time enhanced remarkably the levels of total iso-flavonoid in OCS, showing an increase of 16.5% compared to H_2_O_2_ non-treated chickpea sprouts. Several studies also reported an increase of total iso-flavonoid content in chickpea sprouts, resulting from elicitation under appropriate conditions [3,6]. For example, seed treatment with sodium selenite content (2 mg/100 g seeds) caused a significant increase (83%) of total iso-flavonoids in chickpea sprouts compared to control [3]. Gao et al. [6] reported an 11.5-fold significant increase in the iso-flavonoid concentration of chickpea sprouts as effects of the application of 3% ethanol and 0.03 M saline solution. The variation in the concentrations of isoflavones observed in OCS could be attributed to the PAL enzyme’s increased function in the metabolic pathways of flavonoids [5].

Antioxidant activity of the seed legume has been primarily associated with its phytochemicals, including phenolic, flavonoids, and anthocyanins [48]. It can be seen that the optimal germination condition caused a significant increase in the AOX values obtained by ABTS radical scavenging assay in chickpea sprouts (Table 3). The highest AOX value by ABTS was found in OCS, which was almost 146.1% higher compared to unprocessed chickpea and 14.8% higher compared to H_2_O_2_ non-treated chickpea sprouts. Some researchers have also determined that exogenous elicitors can increase the AOX by ABTS in legume sprouts [21,49,50]. Ampofo and Ngadi [49] reported that the elicitation treated with 300 mM NaCl and 5 mM glutamic acid significantly enhanced the ABTS capacity in common bean sprouts by 13.9% and 22%, respectively, compared to their control sprouts. Liu et al. [22] reported that pea sprouts under exposure to blue LED light increase ABTS values at a rate 1.75 times higher than in control sprouts obtained in darkness. In another study performed by Ampofo and Ngadi [50], the ABTS scavenging capacity of common bean sprouts elicited with ultrasound treatment (360 W, 60 min) was significantly enhanced (32.1%) compared to control. The enlargement of AOX detected in OCS could be attributed to the biochemical metabolism after elicitor treatment, which increased the phenolic compounds [37,42].

## 4. Conclusions

Optimal germination conditions for chickpea seeds treated with hydrogen peroxide (H_2_O_2_) were effectively assessed by employing the RSM and the desirability function. Optimized conditions, 30 mM H_2_O_2_ soaking treatment, and 72 h germination time allowed the production of chickpea sprouts with enhanced levels of phenolic compounds and related antioxidant activity and decreased phytic acid and saponins contents. Optimized germination conditions slightly modified the flavonoid profile in chickpea; eight iso-flavonoids were identified in optimized chickpea sprouts (OCS). This study supports the elicitation with H_2_O_2_ as an effective approach to improve polyphenolics compounds and antioxidant activity by ABTS assay in chickpea sprouts. However, more research is needed to study the antioxidant activities of OCS using different in vitro and in vivo studies to better understand their nutraceutical properties.

## Figures and Tables

**Figure 1 foods-09-01791-f001:**
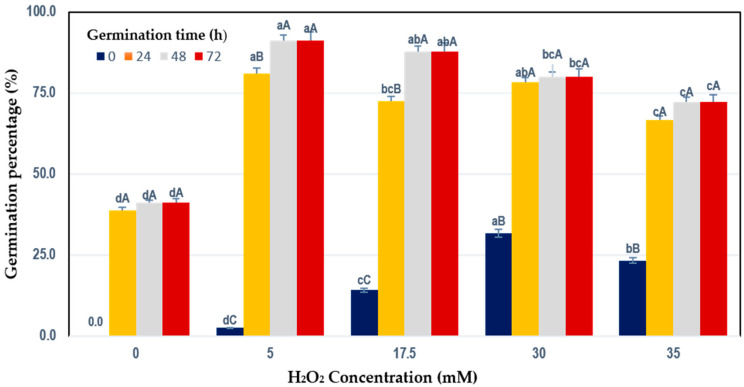
Germination percentage of chickpea seeds elicited with H_2_O_2_ at different concentrations. Values are the mean of three replicates. The same uppercase letter between germination times means no significant difference (*p* < 0.05); the same lowercase letter between H_2_O_2_ concentrations at the same germination time means no significant difference (*p* < 0.05).

**Figure 2 foods-09-01791-f002:**
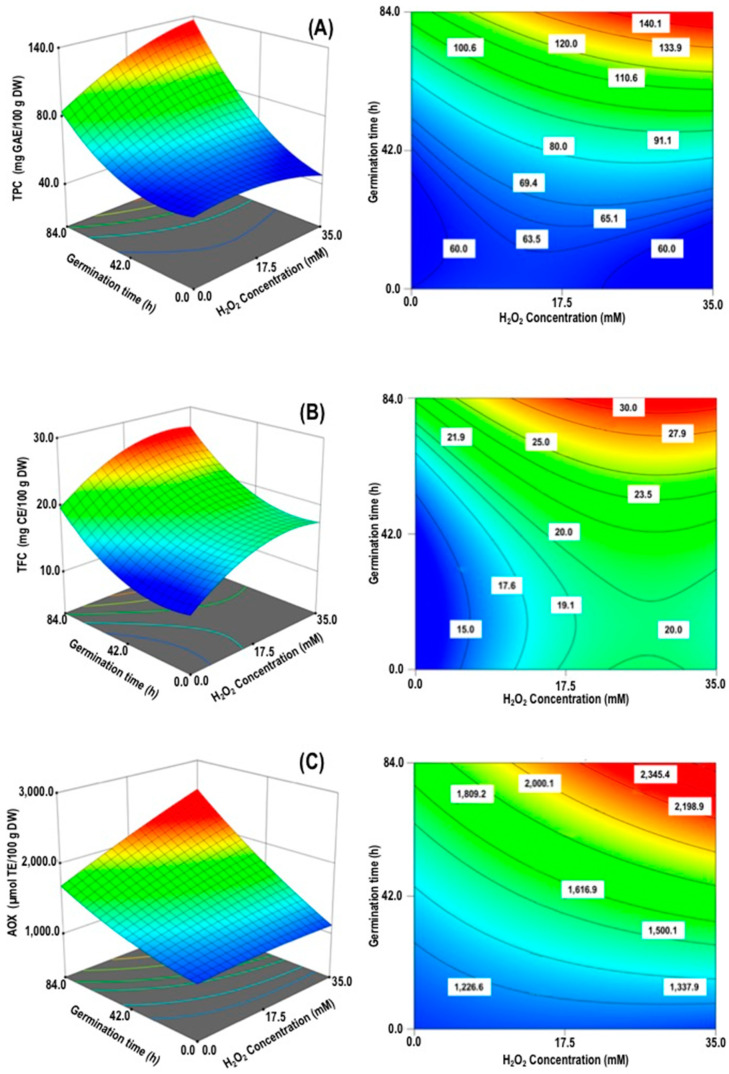
Response surface and contour plots for the effect of the germination condition (H_2_O_2_ concentration and germination time) on the responses variables (**A**) Total phenolic content (TPC); (**B**) Total flavonoids content (TFC); and (**C**) Antioxidant activity (AOX) of chickpea sprouts.

**Figure 3 foods-09-01791-f003:**
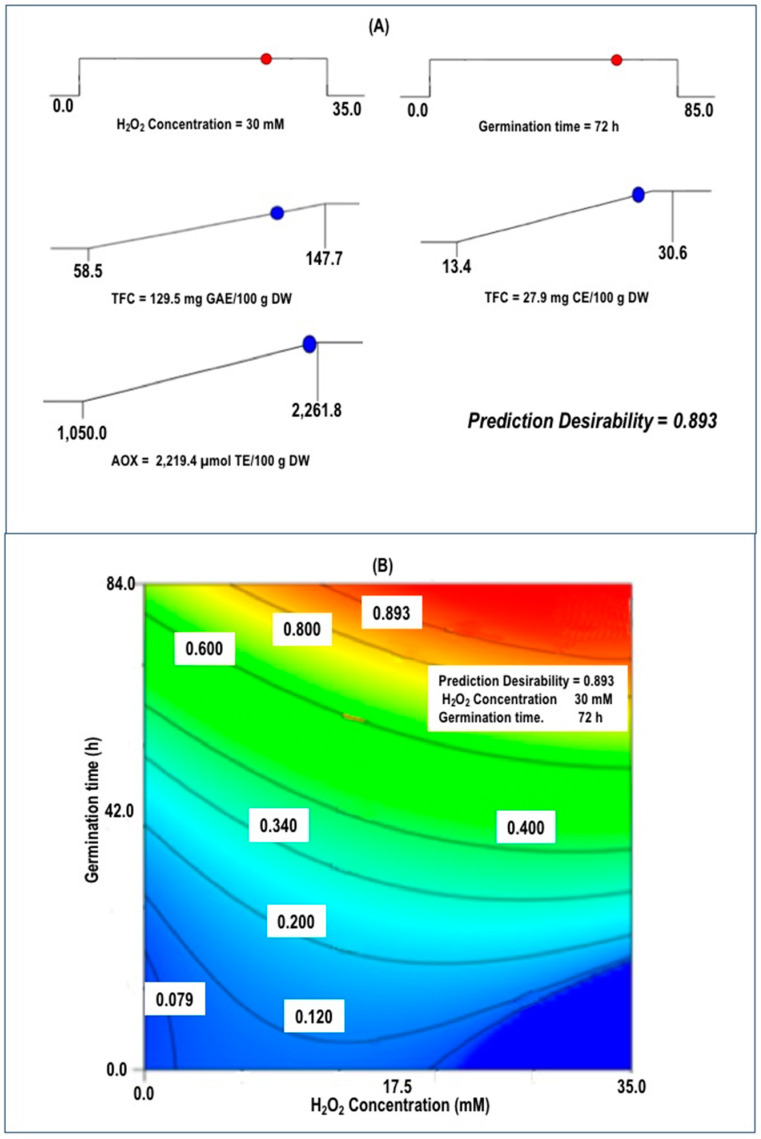
Desirability analysis. (**A**) Plots ramp showing the optimal experimental parameters ((H_2_O_2_) and Gt) that maximize TPC, TFC, and AOX, (**B**) Contour plot of overall desirability showing the best combination of germination condition (H_2_O_2_ concentration and germination time) for producing optimized chickpea sprouts.

**Figure 4 foods-09-01791-f004:**
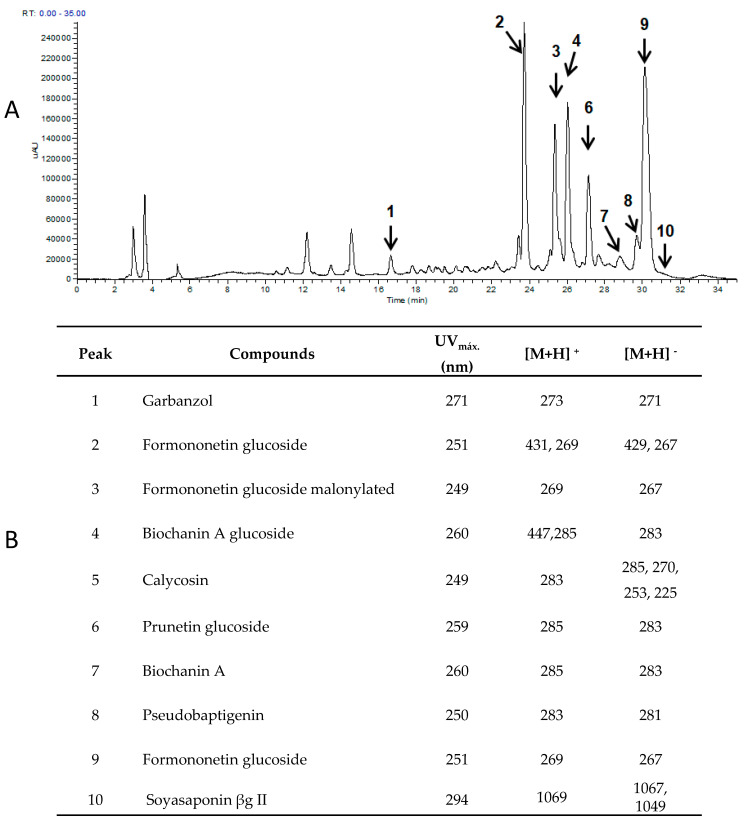
(**A**) chromatogram (260 nm) of iso-flavonoids from optimized chickpea sprouts (OCS) elicited with 30 mM H_2_O_2_ and 72 h germination time. (**B**) Peak assignment of iso-flavonoids in OCS presented according to maximum UV absorption and molecular ions.

**Table 1 foods-09-01791-t001:** Experimental result of response variables evaluated in the chickpea sprouts produced at different combinations of (H_2_O_2_) concentration/germination time.

	Germination Conditions ^1^				
Assay ^2^	Process Variables ^3^		Response Variables ^4^		
	[H_2_O_2_] Concentration (mM) (X_1_)	Germination Time (h) (X_2_)	Total Phenolic Content (TPC) (mg GAE/100 g DW) Y_TPC_	Total Flavonoid Content (TFC) (mg CE/100 g DW) Y_FC_	Antioxidant Activity (AOX) (µmol TE/100 g DW) Y_AOX_
1	5 (−1)	12 (−1)	58.45	13.41	1265.88
2	30 (+1)	12 (−1)	62.61	18.71	1332.59
3	5 (−1)	72 (+1)	89.56	21.23	1629.44
4	30 (+1)	72 (+1)	126.29	27.33	2117.18
5	0 (−1.414)	42 (0)	69.89	13.80	1369.40
6	35 (+1.414)	42 (0)	80.82	22.00	1772.74
7	17.5 (0)	0 (−1.414)	59.55	20.39	942.77
8	17.5 (0)	84 (+1.414)	147.69	30.58	2334.93
9	17.5 (0)	42 (0)	81.94	20.32	1573.07
10	17.5 (0)	42 (0)	76.85	21.92	1473.01
11	17.5 (0)	42 (0)	87.73	20.22	1651.33
12	17.5 (0)	42 (0)	75.54	20.77	1555.78
13	17.5 (0)	42 (0)	83.49	19.78	1529.90

^1^ Central composite design with two factors and five levels; 13 assays. ^2^ Does not correspond to order of experiments. ^3^ H_2_O_2_ concentration (mM), Gt, Germination time (h); values in parentheses are coded levels. ^4^ GAE, Gallic acid equivalent, CE, Catechin equivalent, TE, Trolox equivalent, DW, dry weight basis.

**Table 2 foods-09-01791-t002:** Results of the regression analysis of predicted quadratic polynomial models for each of the response variables of chickpea sprouts.

	Regression Parameter Coefficients		
	Responses Variables ^1^		
Parameter	TPC (mg GAE/100 DW)	TFC (mg CE/100 DW)	AOX (µmol TE/100 g DW)
	**Coded Values**	**Coded Values**	**Coded Values**
Intercept			
β_0_	81.1	+20.6	+1565.1
Linear			
β_1_, H_2_O_2_ concentration (X_1_)	+7.04	+2.87	+155.4
β_2_, Germination time (X_2_)	+27.43	+3.86	+360.2
Quadratic			
β_11_, H_2_O_2_ concentration ($X_{1}^{2}$)	NS	−1.73	NS
β_22_, Germination time ($X_{2}^{2}$)	+9.94	+2.06	NS
Interactive			
β_12_, H_2_O_2_ concentration x Germination time (X_1×2_)	+8.14	NS	+110.2
*p*-value for model	<0.0001	<0.0001	<0.0001
R^2^	0.956	0.967	0.933
R^2^_ajusted_	0.925	0.947	0.886
*p*-value for lack of fit	0.138 ^NS^	0.191 ^NS^	0.726 ^NS^
CV (%)	8.3	5.1	4.8
Adequate precision (PRESS)	17.0	23.8	14.1
**Responses**	**Predicted Models**	**Predicted Values**	**Experimental Values ^2^**
TPC	$Y_{TPC}=81.1+7.04X_{1}+27.43X_{2}+9.94X_{1}^{2}+8.14X_{1}X_{2}$	129.47	138.8 ± 2.01
TFC	$Y_{TFC}=20.6+2.87X_{1}+3.86X_{2}-1,73X_{1}^{2}+2.06X_{2}^{2}$	27.86	27.8 ± 1.32
AOX	$Y_{AOX}=1,565.1+155.4X_{1}+360.2X_{2}+110.2X_{1}X_{2}$	2219.4	2347.2 ± 133

^1^ Total phenolic content (TPC) Total flavonoid content (TFC) Antioxidant activity (AOX); NS, Not significant at *p* < 0.05; ^2^ Mean ± standard deviation of three replicates from experiments to optimal germination condition (30 mM H_2_O_2_, 72 h).

**Table 3 foods-09-01791-t003:** Chemical composition and antinutritional parameters of unprocessed chickpea and chickpea sprouts.

Property	UC	Control	OCS
**Chemical composition (%DW)**			
Protein	15.98 ± 0.15 ^b^	19.99 ± 0.59 ^a^	18.93 ± 0.49 ^a^
Lipids	6.72 ± 0.05 ^a^	7.73 ± 0.12 ^b^	7.96 ± 0.03 ^b^
Ashes	3.02 ± 0.02 ^a^	2.62 ± 0.11 ^b^	2.77 ± 0.01 ^b^
Total carbohydrates	74.27 ± 0.41 ^a^	69.66 ± 0.81 ^b^	71.34 ± 0.92 ^b^
**Antinutrient compounds**			
Phytic acid (mg PAE/100 g DW)	140.03 ± 6.51 ^a^	115.71 ± 4.32 ^b^	97.13 ± 4.11 ^c^
Saponins (mg DE/100 g DW)	426.67 ± 18.1 ^b^	498.68 ± 16.3 ^a^	389.94 ± 12.6 ^c^
Trypsin inhibitors (TIU/mg DW)	6.76 ± 0.18 ^a^	6.24 ± 0.15 ^b^	6.34 ± 0.12 ^b^
**Bioactive compounds**			
Total phenolic (mg GAE/100 g DW)	45.5 ± 1.45 ^c^	128.3 ± 1.01 ^b^	137.8 ± 2.01 ^a^
Total flavonoids (mg CE/100 g DW)	1 0.5 ± 0.81 ^b^	27.6 ± 1.01 ^a^	27.8 ± 1.32 ^a^
Total isoflavones (µg FE/100 g DW)	8.5 ± 1.6 ^c^	221.3 ± 9.2 ^b^	257.9 ± 8.1 ^a^
Antioxidant activity (µ mol TE)/100 DW)	953.9 ± 197 ^c^	2045.1 ± 101 ^b^	2347.2 ± 133 ^a^

Data are expressed as means ± standard deviation of three replicates; values with different letters in the same row show significant difference (*p* < 0.05). UC, unprocessed chickpea; Control, H_2_O_2_ non-treated chickpea sprouts; OCS, optimized chickpea sprouts.

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
