# Peer review of "Improving Polyphenolic Compounds: Antioxidant Activity in Chickpea Sprouts through Elicitation with Hydrogen Peroxide"

_foods, 2020, doi:10.3390/foods9121791_

Round 1
Reviewer 1 Report
The research is of great interest to readers in the food processing and functional food arena. The authors examined a number of different bioactive compounds providing a more complete picture of how fortification using hydrogen peroxide and sprouting can improve the nutritional quality of chickpeas. However, major changes are required to improve the quality and soundness of the manuscript.
- the title is a bit ambiguous, a shorter clearer title would be prefered eg. "Optimizing elicitation of bioactive compounds in chickpea sprouts..."
- there are a number of sentences that needed rephrasing or rewording eg L 45, 58, etc
- the consistency and use of correct tense
- Incorrect use of words eg L45 "enlarged" , in this context phenolic compounds did not grow in size but in quantity thus the correct words would be produced. Also L74 ..was led agreement too
- L47 did not seem to fit
- L52 sentence needs rephasing
- L58 correct the use of tense
(please apply these comments to the reminder of the manuscript)
- Referring to table 1 for experimental design - Table 1 is confusing, I suggest writing your design in section 2.2 itself.
- L97 TPC
- Section 2.7 needs more detail what was the elution profile, solvents used, which compounds had standards for identification or quantification.
- I suggest adding a blind experiment to validate the regression.
Reviewer 2 Report
In overall, the methodology employed to the research is good. However, this manuscript requires extensively revision in English language and grammar. There are several problems in the consistency of terminology (e.g., P-value, TPC, raw seeds vs unprocessed, etc) and the unit used throughout the manuscript.
The introduction requires more extensive review regarding the effect of elicitor on the phenolic content in legume sprout. It would be useful to the readers if the authors could provide more examples of similar study to justify the novelty of this research.
The methodology section is missing statistical analysis. Regarding antioxidant activity, ABTS scavenging activity is a good method to evaluate food antioxidant compounds. However, if it has been used solely in the study, it may not provide sufficient information. Therefore, several antioxidant assays are often employed in the study. It is thus recommended to investigate further using other antioxidant assays.
The results and discussion require further discussion to compare the results in the present study to previously reported values. In section 3.5, have you quantify the individual isoflavonoid component in order to prove that an increased in antioxidant activity in the OCS compared to the 'unprocessed chickpea and control' is linked to these compounds?
Please find more specific comments in the attached file.

Reviewer 3 Report
My comments below :
List abbreviations to minimize comments under tables or in figure legends.
The english style is not homogeneous, particularly at the end of the document.
Abstract :
Correct this sentence : « toward attaining chickpea sprouts was done to for maximizing over 21 response variables such as total phenolic contents (TFC), flavonoid contents (FC),… »
Introduction
l.67 RSM instead of SRM
l.65 à 68 too long sentence. Too many informations. Rewrite please.
Mat & Met
l.97 TPC instead TFC
Be homogeneous in the writting of the equations (l.89 ; 147)
Results
Section 3.1.
Figure 1 should be more commented because there is a lot of information
« Our results showed that between 24 and 72 h after post-imbibition, the germination percentage 169 reached the highest level (80.0-91.1%) during H2O2 concentrations (5 and 30 nM). » Explain like this, I disagree. Please detail more the analysis of figure 1.
Section 3.2.
Reduce the presentation of the RSM results. Either put the equations or table 2 or figure 2, but not the three, these are redondant.
Section 3.5.
« The total protein content from OCS increased by 18.5%, 279 employing an optimal germination combination compared to raw seeds. »
Only the comparison with the control is interesting. Please, delete this sentence and precise that there is no increase of protein content with your optimum.
- 288 OCS instead OGS
l.319 Correct this sentence. English is not correct « The reason for the increased phenolics compounds could be directly related to the action of sprouts through H2O2 stimulates the enzymes in the phenylpropanoid pathway involved for the novo biosynthesis of phytochemical composites »
l.324-328. Be more concise. Add some informations in Mat&Met. English style to correct
l.325 Correct please « through those of genuine flavonoids values ».
l.335 Showed instead shown
l.337 the highest content in isoflavonoids instead the uppermost isoflavonoids content
l.339 were instead was
l.354 seems instead seemed.
Sentence to correct : « The enlargement of AOX on OCS seemed to be attributed to the biochemical metabolism 354 after elicitor treatment, which developed to raise the phenolic composites, for example, isoflavonoids ».
Conclusion
Delete the s in the title
Reduce the section conclusion and make short sentence to be more comprehensible.
Round 2
Reviewer 1 Report
Overall the authors have sufficiently addressed my concerns, however, few grammatical errors still exist mainly in the introduction section.

Reviewer 2 Report
There have been much improvement from previous version in relation to language and consistency of the manuscript. The authors have addressed most of the comments and suggestions. However, there are some points still missing and need to be addressed.
The introduction (Line 56-57), the authors mentioned that "edible sprouts are foods with potential health-promoting properties." Is this sentence true? Has it been proved in in vivo study that consumption of sprouts showed health promoting effect? If so, please provide the reference.
The Materials and Methods, Table 1 shows the experimental result of the response variables. How many replicate of each experiment has been performed? Please provide more detail and also standard deviations if applicable. Please also check the unit of H2O2 concentration and germination time in the table. The unit of individual parameter must be spell out and described at the table footnote.
Section 2.7, the details MS condition and analysis should be rearranged, please see suggestion in the attachment. Please provide more details regarding the standards used for quantification of isoflavonoids. If the individual standard used, how much individual isoflavonoid content in the OCS?
Section 2.9, please recheck and confirm the response variable parameters are correctly described in Line 176-178.
Section 2.10, what is the software used for statistical analysis in this study? please provide detail.
The results and discussion, Section 3.5, the values of chemical composition described in the text are not in agreement to the values presented in the Table 4.
Figure 3, please check and change the concentration unit of H2O2 if applicable.
Table 4, the unit content of all parameters must be presented full, i.e., 'mg' PAE/100 g DW
Line 367-368, please remove unnecessary parentheses.
Line 388, further discussion is required to compare the antioxidant activity of the present study to other studies.
Conclusion, in order to prove the claim made here that "The elicitation using H2O2 improved the antioxidant activity", further investigation of antioxidant activity using other approaches is recommended. Only ABTS value is not sufficient to provide conclusive outcomes as 'antioxidant activity', unless, the statement must be changes to "... improve ABTS value" or provide suggestion to the reader that "further investigation is required using different antioxidant assay".
Other specific comments regarding language, formatting and consistency of terminology and unit used are provided in the document attached.
